# BRAF-Mutated Colorectal Cancer: Clinical and Molecular Insights

**DOI:** 10.3390/ijms20215369

**Published:** 2019-10-28

**Authors:** Francesco Caputo, Chiara Santini, Camilla Bardasi, Krisida Cerma, Andrea Casadei-Gardini, Andrea Spallanzani, Kalliopi Andrikou, Stefano Cascinu, Fabio Gelsomino

**Affiliations:** 1Department of Oncology and Hematology, Division of Oncology, University of Modena and Reggio Emilia, 41121 Modena, Italy; francesco1990.caputo@libero.it (F.C.); santinic_90@libero.it (C.S.); camilla.bardasi@gmail.com (C.B.); kridi90@gmail.com (K.C.); andrea.spallanzani@gmail.com (A.S.); k.andrikou@hotmail.com (K.A.); cascinu.stefano@hsr.it (S.C.); fabiogelsomino83@yahoo.it (F.G.); 2IRCCS San Raffaele Scientific Institute Hospital, 20019 Milan, Italy

**Keywords:** colorectal cancer, BRAF mutation, BRAF inhibitors, molecular targets, targeted therapy

## Abstract

Colorectal cancer (CRC) is one of the leading causes of mortality and morbidity in the world. It is a heterogeneous disease, which can be classified into different subtypes, characterized by specific molecular and morphological alterations. In this context, BRAF mutations are found in about 10% of CRC patients and define a particular subtype, characterized by a dismal prognosis, with a median survival of less than 12 months. Chemotherapy plus bevacizumab is the current standard therapy in first-line treatment of BRAF-mutated metastatic CRC (mCRC), with triplet (FOLFOXIRI) plus bevacizumab as a valid option in patients with a good performance status. BRAF inhibitors are not so effective as compared to melanoma, because of various resistance mechanisms. However, the recently published results of the BEACON trial will establish a new standard of care in this setting. This review provides insights into the molecular underpinnings underlying the resistance to standard treatment of BRAF-mutated CRCs, with a focus on their molecular heterogeneity and on the research perspectives both from a translational and a clinical point of view.

## 1. Introduction

Colorectal cancer (CRC) is one of the leading causes of mortality and morbidity in the world. With approximately 1,849,518 new cases estimated and 880,792 deaths per year, it also represents the third most common cancer worldwide and the second cause of cancer-related mortality, after lung cancer. In terms of geographical distribution, CRC incidence and prevalence have risen in industrialized countries [1]. However, in recent years the incidence and mortality rates of CRC have grown higher in Eastern Europe, Latin America, and Asia than other countries.

While many cases of CRC are diagnosed at an early stage and treated with curative surgery, a large number of patients develop synchronous or metachronous metastatic disease with a five-year survival rate of roughly 13% [2].

Historically, the treatment of metastatic CRC (mCRC) has been based on the combination of cytotoxic agents, such as irinotecan or oxaliplatin, associated with 5-FU and leucovorin or capecitabine (FOLFIRI/FOLFOX/FOLFOXIRI or CAPIRI/CAPOX regimens), which resulted in an average survival of 18 months [3].

In the last decades, the approval of targeted therapies like anti-epidermal growth factor receptor (EGFR) and anti-vascular endothelial growth factor (VEGF) antibodies has unleashed a revolution in the treatment landscape of mCRC, with overall survival (OS) approaching 30 months in modern clinical trials [4,5].

Recent findings in molecular biology demonstrated that mCRC is not a homogenous disease, but it can be classified into different subtypes, which are characterized by specific molecular and morphological alterations.

Mutations in the BRAF gene are examples of such oncogenic events and are found in about 10% of CRC patients [6]. These mutations are associated with the female gender, often right-sided, advanced stage, mucinous histology, defective mismatch repair (dMMR), and a serrated adenoma pathway. Furthermore, BRAF-mutated CRCs are characterized by a dismal prognosis and resistance to standard therapies, with a median OS (mOS) of approximately 12 months [7].

This review provides insights into the molecular underpinnings underlying the resistance to standard treatment of BRAF-mutated CRCs, with a focus on their molecular heterogeneity and on the research perspectives both from a translational and clinical point of view.

### 1.1. Tumor Heterogeneity in Colorectal Cancer

CRC is characterized by different pathways of carcinogenesis and represents a heterogeneous disease with different molecular landscapes that reflect histopathological and clinical information.

Two molecular pathological classification systems for CRC have recently been proposed. The first is the integrated molecular analysis by The Cancer Genome Atlas project (TCGA) [8], based on a wide range of genomic and transcriptomic characterization studies of CRC using array-based and sequencing technologies. This approach classifies CRC into three major groups which are consistent with previous classification systems:1)hypermutated cancers (~16%) with microsatellite instability (MSI) due to defective mismatch repair (∼13%) usually caused by MLH1 silencing via promoter hypermethylation;2)ultramutated cancers (~3%) with DNA polymerase epsilon or delta 1 (POLE or POLD1) exonuclease domain (proofreading) mutations (EDM), with the malfunctioning enzyme introducing incorrect nucleotides during DNA replication; and3)chromosomal instability (CIN; ~84%) with a high frequency of DNA somatic copy number alterations, usually arising as a consequence of a combination of oncogene activation (e.g., KRAS, PIK3CA) and tumor suppressor gene inactivation (e.g., APC, SMAD4, and TP53) by allelic loss and mutation, which go along with changes in tumor characteristics in the adenoma-to-carcinoma sequence.

The second classification was proposed by the Consensus Molecular Subtypes (CMS) Consortium [9] which, by analyzing CRC expression profiling data from multiple studies, described four CMS groups associated with different patients’ outcomes: the first category CMS1 (MSI-immune, 14%) included CRC hypermutated due to dMMR with MSI and MLH1 silencing and accordingly CpG island hypermethylation phenotype-high (CIMP-high) with frequent BRAF mutations, and a low number of somatic copy number alterations (SCNAs). The remaining microsatellite stable (MSS) cancers are subcategorized into three groups: CMS2 (canonical, 37%), CMS3 (metabolic, 13%), and CMS4 (mesenchymal, 23%), with a residual unclassified group (mixed features, 13%).

### 1.2. BRAF Mutations

BRAF is a serine/threonine protein kinase involved in the signaling cascade—known as the mitogen-activated protein kinase (MAPK) pathway—that drives cell proliferation, differentiation, migration, survival, and angiogenesis. Dysregulation of this pathway underlies many instances of tumorigenesis. This pathway is composed of the RAS small guanidine triphosphatase (GTPase), which activates the RAF family proteins (ARAF, BRAF, and CRAF, also known as RAF1). Activated RAF proteins lead to phosphorylation and activation of MEK1/2 proteins, which subsequently phosphorylate and activate ERKs. ERKs phosphorylate a variety of substrates, including multiple transcription factors and regulate several key cellular activities (Figure 1).

Approximately 96% of all BRAF mutations are a T1799A transversion in exon 15, which results in a valine amino acid substitution: V600E [10]. This appears to mimic regulatory phosphorylation, increasing BRAF activity approximately 10-fold as compared to the wild-type (WT) counterpart. Interestingly, BRAF-mutated mCRC tumors emerge in CRC pathogenesis as a distinct biological entity and characterized by a clinical and molecular heterogeneity [8].

In fact, patients with BRAF V600E mutation segregate into two different gene expression subtypes with distinct molecular patterns and different potential therapeutic targets [11]:-subtype 1 (BM1) is highly active in KRAS/mTOR/AKT/4EBP1 signaling and in genes associated with macrophage infiltration and epithelial–mesenchymal transition (EMT). This group is associated with poorer survival. BM1 shows an overall stronger immune profile emphasized by activation of pathways such as IL2/STAT3, TNF alfa signaling via NF-κB, IL6/JAK/STAT3, and allograft rejection. The pathway analyses therefore indicate that BM1 is characterized by mTOR and RAS signaling and high score apoptosis signatures (for example protein BH3, the only pro-apoptotic BIM protein). Moreover, BM1 presents a higher inflammatory response, correlated with the different expression of protein like SYK that transmits signals in B and T cells and STAT5α, whose expression correlates with immune activation. Surprisingly, CMS-4 includes BRAF mutants BM1.-subtype 2 (BM2) is active in cell-cycle checkpoint associated genes. Notably, trends are found for an improved prognosis and association with MSI with BM2 subtype. Moreover, the mTORC1 signature (complex activation can lead to 4EBP1 and S6K pathway activation) is mostly detected in BM2, which confirms a major involvement in the metabolic process.

## 2. Clinical Data on BRAF V600E- and Non-V600-Mutated CRC

Owing to the rarity of BRAF mutations [12], clinical data about this poorly represented subpopulation of CRCs are scanty. However, based on the available data, a significant heterogeneity of BRAF mutations has emerged.

Indeed, recent studies identified two different molecular phenotypes of CRC based on BRAF mutation status: BRAF V600E- and non-V600-mutated CRC [13]. These two molecular subtypes of CRC are characterized by distinct clinical and pathological features.

In fact, as emerged in a recent meta-analysis [14], BRAF V600E-mutated tumors more commonly arise from serrated adenomas, mainly in the right colon, with a higher incidence in women and elderly patients (age >60-years-old). Moreover, BRAF V600E-mutated CRCs are often poorly differentiated and present a mucinous histotype. Interestingly, a mutually exclusion relationship between KRAS mutation and BRAF V600E (only 0.56% were both KRAS and BRAF mutated) and between BRAF V600E and TP 53 mutations has been reported. Furthermore, a strong positive correlation between BRAF V600E and dMMR (46.15% of all MSI-high (H) resulted BRAF V600E- mutated) has been demonstrated [15], whereas BRAF V600E is negatively associated with CIMP. On the contrary, non-V600 mutations (which account for about 2.2% of mCRC patients and affect codons 594 and 596) represent a distinct population in terms of age, gender, histology, MSI status, metastatic sites, and prognosis [13,16]. Indeed, they are found more often in males and younger than the BRAF V600E-mutated subgroup; they often present with low-medium grade histological tumors, located in the left-sided colon whereas peritoneal involvement in advanced disease stage is rare [13,16]. Furthermore, BRAF non-V600E-mutated CRCs exhibit a completely inverted molecular behavior, often associated with concomitant RAS mutations and rarely presenting dMMR. This suggests that, at least in some cases, rare BRAF mutations may confer a less proliferative advantage to cancer cells as compared to other mutations with a negative prognostic impact. Surprisingly, changes induced by BRAF 594 and 596 mutations are able to phosphorylate ERK nearly two-fold more than BRAF-WT, but only in the presence of endogenous CRAF. This finding led to the discovery of the mechanisms involving BRAF-CRAF dimerization.

Based on preclinical data, a further classification of BRAF mutations has been proposed:RAS-independent BRAF mutations signaling as monomers (class 1)RAS-independent BRAF mutations signaling as dimers (class 2)RAS-dependent BRAF mutations with impaired kinase activity or kinase-dead (class 3) [17].

BRAF V600E mutation belongs to class 1, while BRAF 601 and 597 to class 2, and BRAF 594 and 596 to class 3. Schirripa et al. [18] demonstrated a correlation between this classification and the clinical behavior of distinct subtypes. Indeed, class 3 subtypes were more frequently left-sided, node-negative, with no peritoneal metastasis compared to class 1, whereas class 2 was similar to class 1. In terms of survival, mOS was 21.0 vs. 23.4 vs. 44.5 vs. 42.2 months in BRAF-mutated class 1, 2, 3, and BRAF-WT (*p* < 0.0001), respectively. These data confirmed the findings reported by Jones et al. [13]. They showed important differences in terms of prognosis between BRAF V600 and non-V600 CRCs (>50% class 3), with a substantially longer mOS of 60.7 months in BRAF non-V600E-mutated patients, compared to 11.4 months in BRAF V600E-mutated, but also compared to the 43.0 months of BRAF-WT population, emphasizing the less aggressive behavior of the BRAF non-V600E-mutated CRCs.

### 2.1. BRAF Mutations as a Prognostic Factor

Several clinical studies have been conducted aiming at defining the role of BRAF mutations as a potential prognostic biomarker in CRC patients. Current available data derive mainly from patients presenting BRAF V600E mutations, being the most common variant. Regardless of disease stage, the presence of this mutation appears to be correlated with greater chemoresistance and worse prognosis [19].

In this respect, Farina-Sarasqueta et al. showed that BRAF V600E mutation is an independent negative prognostic factor for survival in stage II–III CRCs (HR 0.45, 95% confidence interval (CI) 0.25–0.8), while it does not seem to influence disease-free survival (DFS) [20]. Similar conclusions were reported by a retrospective analysis of three randomized trials [21]. These data demonstrate that patients with BRAF V600E-mutated CRC have a similar probability of relapse compared to BRAF-WT, but a significantly shorter post-relapse survival. As previously reported, BRAF V600E-mutated CRCs frequently present MSI, which is considered to be a good prognostic factor in early-stage CRCs. Indeed, MSI-H CRC patients without the BRAF mutation demonstrated the best prognosis, while MSS/BRAF V600E patients exhibited the worst; MSS/BRAF-WT and MSI/BRAF V600E CRCs seems to have an intermediate prognosis [22,23]. Interestingly, in the post-hoc analysis of the PETACC-8 trial [24], it was reported that in the MSI-H subpopulation, the presence of BRAF V600E mutation was associated with longer DFS as compared to BRAF-WT patients, but there was no effect on OS (DFS: HR 0.23, 95% CI 0.06–0.92, *p* = 0.04; OS: HR 0.19, 95% CI 0.03–1.24, *p* = 0.08), suggesting that MSI-H is a protective factor against BRAF mutation in early-stage CRC. Similar results were reported by Seppala et al. [25]. However, other studies reported no impact of BRAF mutation on MSI-H early-stage CRCs [26]. Therefore, based on these data, BRAF V600E mutation can be considered an independent negative prognostic factor in early-stage MSS CRC, while its role in the MSI-H subpopulation remains controversial.

The negative impact of the BRAF mutation has also been reported in advanced-stage CRC. In the AIO KRK0207 trial, BRAF mutation was reported as the strongest unfavorable prognostic factor (HR 3.16; 95% CI 2.17–4.60; *p* < 0.0001) compared to RAS status and primary tumor location [27]. In the prognostic analysis of the FOCUS trial [28] there was no evidence that the BRAF mutation alone had an effect on progression free survival (PFS) (HR 1.14; 95% CI 0.86–1.52; *p* = 0.37), but it seemed to have a relevant impact on OS (HR 1.82; 95% CI 1.36–2.43; *p* < 0.0001), describing a similar behavior to early-stage disease. However, in a pooled analysis of CAIRO, CAIRO2, COIN, and FOCUS trials [29], BRAF mutation had a negative impact on both PFS (6.2 vs. 7.7 months; HR 1.34; 95% CI 1.17–1.54; *p* < 0.001) and OS (11.4 vs. 17.2 months; HR 1.91; 95% CI 1.66–2.19; *p* < 0.001). Similar results were also seen in three different other studies [30,31,32]. Specifically, Innocenti et al. reported the aggressive behavior of BRAF V600E-mutated tumors with a mOS of 13.5 months compared to 30.6 months for patients with BRAF-WT tumors. Moreover, based on analysis of three large randomized trials [33], it was suggested that outcomes markedly diverge between BRAF-mutated and BRAF-WT CRC after first-line progression.

Considering MMR status in advanced-stage CRC, BRAF mutation seems to have a negative prognostic role on PFS and OS only in MSS CRC patients, whereas there is no difference in terms of survival between MSI-H/BRAF-mutated and MSI-H/BRAF-WT cancers [29].

The impact of the BRAF mutation remains negative also in post-metastasectomy settings. BRAF-mutated CRC showed worse survival compared to BRAF-WT ones after lung [34] and liver [35,36] metastasectomies. Indeed, in the largest series published by the Mayo Clinic of 21 patients who underwent resection of liver metastases from BRAF-mutated CRC, the mean progression free survival (mPFS) and mOS were longer than in the non-metastasectomy cohort (13.6 and 29.1 months vs. 6.2 and 22.7, respectively), with one patient who remained relapse-free for more than two years. In multivariate analysis, metastasectomy remained significant for improved survival outcomes (HR 0.52; 95% CI 0.07–1.02; *p* = 0.02) [37]. Although BRAF V600E mutations portend a dismal prognosis, exceptions to this general rule do exist, therefore exclusion from resection of metastatic disease should not be solely based on the presence of BRAF mutations.

Emerging data confirm that tumor microenvironment can impact disease prognosis. It has been demonstrated that chronic inflammation with a high expression of COX2-PGE2 plays a fundamental role in the genesis of CRC [38], as PGE2 suppress the anti-tumor effect deployed by the immune system. BRAF mutation determines an upregulation of the RAF-MAPK pathway, which leads to downstream activation of PTGS2 (COX-2) and to the increase of PGE2 production [39]; this leads to higher survival and faster replication of tumor cells. In a study carried out by Kosumi and colleagues, considering BRAF-mutated CRC, the subgroup with higher COX-2 expression presented lower disease-specific survival (DSS) (HR 2.44; 95% CI 1.39–4.28); the association between COX2 and worse survival did not reach statistical significance in the BRAF-WT population (HR 0.82; 95% CI 0.65–1.04). As an explanation of this mechanism, the authors proposed that PGE2 accumulation in the tumor microenvironment resulted in greater resistance to the local immune system [40].

Recently Loupakis et al. elaborated a prognostic score in order to better stratify BRAF V600E-mutated CRC patients. Considering main clinical and pathological aspects related to patients and specifically to the disease, BRAF V600E-mutated CRCs have been divided into three groups: high, intermediate, and low risk, with mOS expectations of 6.6, 15.5, and 29.6 months, respectively. They elaborated a valid and easy-to-perform scoring system which could be useful in clinical practice and stratified patients enrolled in clinical trials [41]. For example, this tool could be useful to decide which patients with liver metastases from BRAF V600-mutated CRC should undergo resection and which should not.

### 2.2. BRAF as a Predictive Factor

The therapeutic approach to BRAF-mutated CRC has always been considered challenging, given its intrinsic resistance to chemotherapy. Therefore, intensive strategies with drugs combination have been tested. The triplet FOLFOXIRI in association with bevacizumab initially produced favorable results in a phase II validation trial for the subgroup of BRAF-mutated CRC, with a significant gain in terms of survival (mPFS and mOS of 11.8 and 24.1 months, respectively) [42]. The subsequent phase III TRIBE trial demonstrated a survival advantage with FOLFOXIRI plus bevacizumab as compared to the FOLFIRI plus bevacizumab. Of note, in the subgroup of BRAF-mutated CRC there was a clinically relevant advantage, although not statistically significant, probably due to the small number of patients (mOS 10.7 vs. 19.0 months, *p* = 0.54; mPFS 5.5 vs. 7.5 months, *p* = 0.57, for doublet vs. triplet, respectively) [43]. Therefore FOLFOXIRI + bevacizumab became the preferable first-line treatment for these patients.

Although the use of anti-VEGF has become a therapeutic standard, the impact of antiangiogenic drugs has not been statistically demonstrated in BRAF-mutated patients. In a post-hoc analysis of the AVF2107g [44] and AGITG MAX [45] trials, BRAF gene mutation status was prognostic for OS but not predictive of an outcome with bevacizumab, although data showed a numerical improvement in survival. Analogously, in the VELOUR trial the impact of the addition of aflibercept to FOLFIRI in second-line treatment of CRC was tested. In the survival analysis stratified by prognostic biomarkers, the BRAF-mutated subgroup had a relevant improvement in OS compared to the BRAF-WT population, although data did not reach statistical significance (HR 0.49; 95% CI 0.22–1.09; *p* = 0.08) [46]. Similar results have been reported with ramucirumab in the RAISE trial [47].

Acknowledging that the results may be skewed by the limited number of CRC patients with BRAF mutation, controversial data have been published about the impact of anti-EGFR drugs on these patients, with some data suggesting a potential negative predictive role of BRAF mutations [48]. As of today, the use of anti-EGFR cetuximab and panitumumab in BRAF-mutated/RAS WT CRC has no therapeutic restrictions. The addition of anti-EGFR to FOLFOX (PRIME trial) [49] or FOLFIRI (PICCOLO trial) [50] did not demonstrate a clinical benefit, with a detrimental effect of anti-EGFR therapy in BRAF-mutated CRC in the PICCOLO trial (OS: HR 1.84; 95% CI 1.10–3.08; *p* = 0.029). However, survival curves of the CRYSTAL study showed an improving trend in both PFS (8.0 vs. 5.6 months, *p* = 0.87) and OS (14.1 vs. 10.3 months, *p* = 0.74), but statistical significance was not achieved because of the small number of patients involved [51]. A pooled analysis of this study along with the OPUS trial confirmed a better objective response rate (ORR), PFS, and OS in patients with BRAF mutation, suggesting that BRAF mutation does not confer resistance to anti-EGFR drugs [52]. Two different meta-analyses with conflicting results were conducted in order to clarify this issue [53,54]; therefore the predictive role of BRAF mutations in patients with CRC treated with anti-EGFR remains a matter of debate. Recently Geissler et al. presented results from the VOLFI trial (AIO-KRK0109), in which FOLFOXIRI was compared to FOLFOXIRI + anti-EGFR (panitumumab); the first-line treatment with mFOLFOXIRI + panitumumab resulted in a significantly higher ORR compared to FOLFOXIRI alone (ORR 85.7% vs. 60.6% *p* = 0.0096) while there was no difference in PFS between both arms. Best results were obtained in symptomatic patients and BRAF-mutated mCRC; an outstanding ORR of 71% in BRAF-mutated CRC vs. 22% in BRAF-WT was observed, although these data did not reach statistical significance, again due to the small number of patients [55]. The interest in BRAF non-V600E mutation has increased, because of its different behavior as compared to BRAF V600E-mutated CRC. Recently, Johnson et al. characterized the treatment efficacy of anti-EGFR in these peculiar molecular subtypes [56]: according to Yao’s classification of the BRAF mutation pathway, no responses to anti-EGFR therapy were reported in class II and III (BRAF non-V600E mutation), suggesting that the effect of cetuximab and panitumumab is limited to class I. Prospective trials are needed to better clarify the clinical behavior of these atypical BRAF mutations.

## 3. Resistance of BRAF-Mutated CRC to BRAF Inhibitors

Recently, BRAF inhibitors (iBRAF), such as vemurafenib, dabrafenib and encorafenib, have revolutionized the treatment of BRAF V600E metastatic melanoma in monotherapy or in combination with other drugs [57,58,59]. Furthermore, they demonstrated promising activity also in BRAF V600E-mutated non-small cell lung cancer, anaplastic thyroid cancer, cholangiocarcinoma, Erdheim–Chester disease, and Langerhans cell histiocytosis [60].

Additionally, single-agent vemurafenib was shown to arrest cell proliferation and inhibit tumor growth in CRC cell lines and xenograft models expressing BRAF V600E mutation, respectively [61]. For this reason, Kopetz et al. led a phase II study evaluating vemurafenib in patients with previously-treated BRAF-mutated mCRC. Among the 21 patients enrolled, only one reached a partial response, while seven reported to have stable disease, with mOS and mPFS of 7.7 months and 2.1 months, respectively [62]. These results are clearly disappointing, especially if compared to the remarkable success gained in melanoma. Other studies confirmed the lack of efficacy of iBRAF in BRAF-mutated mCRC [60,63,64].

### 3.1. Why the Results Are so Different

It is well-known that the molecular landscape of BRAF V600E-mutated mCRC is more complex and heterogeneous as compared to melanoma. In mCRC, MAPK activity is normally driven by mutated BRAF, and the RTK-mediated activation of RAS is restricted by ERK negative feedback signals. Instead, iBRAF treatment leads to reactivation of MAPK pathways by triggering significant adaptive feedback signaling networks. In fact, iBRAF causes an initial decrease of MAPK signaling, leading to a loss of expression of ERK negative feedback. This implicates an increase of RTK-mediated RAS activation and the recruitment of other RAF kinases, which produce RAF dimers (such as CRAF), restoring MAPK pathways signaling [65] (Figure 2). Significantly, RTK signaling is present to a higher level in CRC than in melanoma, and one of them (EGFR) is mainly responsible for MAPK reactivation in BRAF-mutated mCRC, as shown in preclinical models [65,66].

Other pathways may be involved in this kind of resistance, such as PI3K/AKT pathway. It has been shown that BRAF-mutated mCRC cell lines have higher activation of several proteins in this pathway as compared to melanoma cell lines, and that mCRC cell lines with mutations in PI3K/AKT pathway or loss of PTEN are more resistant to growth inhibition by BRAF inhibitors as compared to cell lines without these alterations [67].

Also, Wnt pathway plays a role in promoting resistance to iBRAF. In a murine model of BRAF V600E-induced colon tumorigenesis, it has been described that Wnt pathway activation promotes the progression from intestinal hyperplasia to carcinoma [68]. Furthermore, alterations in this pathway have been identified more frequently in BRAF V600E-mutated mCRC cells [9]. On one hand, there is a close relationship between the Wnt and MAPK pathways, with ERK activating the Wnt co-receptor LRP6 and regulating the transcription of the proto-oncogene c-Myc [69,70]. On the other hand, the use of iBRAF upregulates this pathway regardless of MAPK signaling through FAK (focal adhesion kinase) activation, potentially representing an alternative pathway of tumor development [71].

### 3.2. Strategies to Overcome Resistance to BRAF Inhibitors

Based on these findings, many clinical trials were designed and conducted with the aim of overcoming resistance to iBRAF, using a combination of targeted therapies and/or chemotherapy.

#### 3.2.1. Doublet Combinations

As previously mentioned, in BRAF-mutated mCRC MAPK reactivation is regulated by RTK signaling, especially EGFR. For this purpose, in a phase II “basket” trial, the combination of vemurafenib and cetuximab was evaluated in 27 patients with BRAF V600E-mutated mCRC. One patient had a partial response and 69% had stable disease with mOS and mPFS of 7.1 and 3.7 months, respectively [60]. In another trial, the combination of vemurafenib and panitumumab was investigated in 15 pre-treated patients with BRAF V600E-mutated mCRC. Two patients had a partial response and six had stable disease [72]. Similarly, other combinations such as vemurafenib and erlotinib, encorafenib and cetuximab, dabrafenib and panitumumab, have been evaluated in other trials with response rates ranging from 4% to 23% [73,74,75].

Given the revolutionary impact of the combination of BRAF and MEK inhibitors in the clinical management of melanoma, it was also evaluated in BRAF-mutated mCRC by Corcoran et al. In total, 43 patients were treated with dabrafenib and trametinib, with 56% of stable disease and an ORR of 12% [76].

The mentioned trials showed a partial activity of all these combinations, although far away from the impressive results in melanoma.

#### 3.2.2. Triplet Combinations

In order to improve the clinical outcomes of these patients, triplet drug combinations have also been evaluated. In a trial by Corcoran et al., three cohorts were treated with dabrafenib plus panitumumab, dabrafenib plus trametinib plus panitumumab, and trametinib plus panitumumab, respectively. The authors showed an ORR for triplet therapy of 21%, better than doublet therapy, but with an increase of adverse events, in terms of diarrhea and skin toxicity (rash and dermatitis acneiform); mPFS was 4.1 months, while mOS reported was 9.1 months [77].

As BRAF inhibition can induce PI3K pathway modulation, PI3K inhibitors were also explored. In a phase Ib dose-escalation study, 28 refractory BRAF-mutated CRC patients were included in two arms: encorafenib and cetuximab vs. encorafenib, cetuximab and alpelisib, an inhibitor of alpha subunit of PI3K. An ORR of 18% and a disease control rate (DCR) of 93% were reported for the triplet therapy [78]. For this reason, a phase II trial was performed, in which 52 patients were treated with the triplet therapy [74]. At an interim analysis, mPFS was 5.4 months vs. 4.2 months for the doublet therapy. Also, in this case toxicity was higher with the triplet, including anemia, hyperglycemia, and increased lipase.

Other combinations were tested, including chemotherapy. Based on preclinical data showing a great antitumor activity by using doublet or triplet therapy, a phase 1 trial, combining vemurafenib, cetuximab, and irinotecan was developed. A total of 18 BRAF-mutated CRC patients were included, with an ORR of 35% and mPFS of 7.7 months [79]. The following randomized phase II trial (SWOG 1406) combining irinotecan and cetuximab with or without vemurafenib included 106 patients. mPFS was 4.3 months in vemurafenib arm vs. 2.0 months of the control arm; in addition, response and disease control rates were higher in the vemurafenib arm [80]. Recently, the results of the phase III BEACON trial were published, in which 665 patients with pre-treated BRAF-mutated mCRC were randomized to receive encorafenib, cetuximab, and binimetinib (a MEK inhibitor) vs. encorafenib and cetuximab vs. irinotecan/FOLFIRI and cetuximab. mOS was 9.0 months for the triplet combination vs. 5.4 months for standard therapy and 8.4 months for doublet. In the triplet and doublet arms the ORR was 26% and 20%, while mPFS was 4.3 months and 4.2 months, respectively [81]. As compared to the previously mentioned studies, this is the largest cohort ever studied and the first phase III trial to demonstrate a survival and response advantage in the setting of pre-treated BRAF-mutated CRCs.

Table 1 summarizes the main clinical trials investigating iBRAF treatments for BRAF-mutated mCRC.

## 4. Future Perspectives

The treatment of BRAF-mutated CRC had a rapid evolution in the last years and relevant changes are expected in the near future. In the setting of second- and third-line treatments, the triple combination of BEACON trial will probably represent a ‘game changer’.

An ongoing study (ANCHOR-CRC) is investigating the effects of the same triplet therapy as a first-line treatment for patients with BRAF-mutated CRC [83].

Other potential targets include the Wnt/beta-catenin pathway. The results of the trial NCT02278133, which enrolls patients with both BRAF-V600E and Wnt pathway mutations and treated with the Wnt inhibitor WNT974 in combination with encorafenib and cetuximab, are expected [84].

Furthermore, immunotherapy (such as immune-checkpoint inhibitors like pembrolizumab or nivolumab) could play an important role in this particular setting, given the well-recognized association between BRAF-V600E mutations and MSI [15]. In the phase II study CHECKMATE 142 treatment with nivolumab associated or not with ipilimumab (a CTLA-4 inhibitor) was evaluated in MSI-high/dMMR CRC. A total of 12 among the 24 enrolled patients receiving nivolumab harbored BRAF-V600E mutation, with an ORR of 25%, and of 55% for combination immunotherapy. Consequently, the possibility of durable disease control may be even much higher with checkpoint inhibitors [85,86]. An overview of the main ongoing studies in BRAF-mutated mCRCs is provided in Table 2.

## 5. Conclusions

BRAF V600E mutations are present in 7%–10% of CRCs and define a particular subtype characterized by a dismal prognosis. Chemotherapy plus bevacizumab is the current standard therapy in first-line treatment of BRAF-mutated mCRC. An upfront treatment with a triplet (FOLFOXIRI) plus bevacizumab may represent a valid option in patients with good performance status. BRAF inhibitors are not as effective as melanoma because of different resistance mechanisms. Nevertheless, many trials investigated the role of iBRAF in association with EGFR, MEK, and PI3K inhibitors, particularly in second-line therapy and beyond, highlighting only a partial activity. The unprecedented results of the BEACON trial established a new standard of care in this setting of patients. While immunotherapy will be an option, at least in MSI-H CRCs, ongoing research will hopefully demonstrate if combination strategies with iBRAF and other drugs can hold the promise seen in preclinical studies.

## Figures and Tables

**Figure 1 ijms-20-05369-f001:**
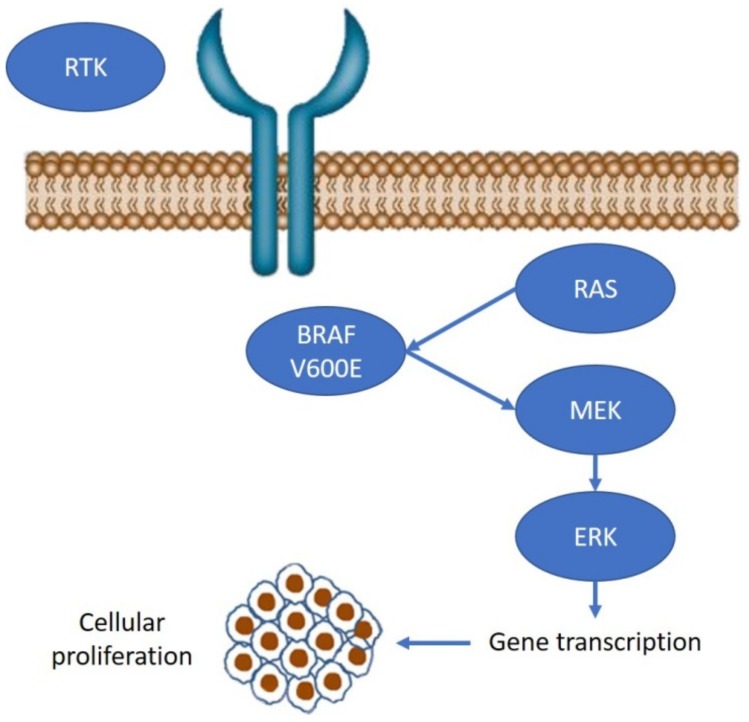
Mitogen-activated protein kinase (MAPK) pathway in BRAF V600E-mutated metastatic colorectal cancer (mCRC). RAS activates the RAF family proteins (ARAF, BRAF, and CRAF). Activated RAF proteins lead to phosphorylation and activation of MEK1/2 proteins, which subsequently phosphorylate and activate ERKs, leading to cell growth.

**Figure 2 ijms-20-05369-f002:**
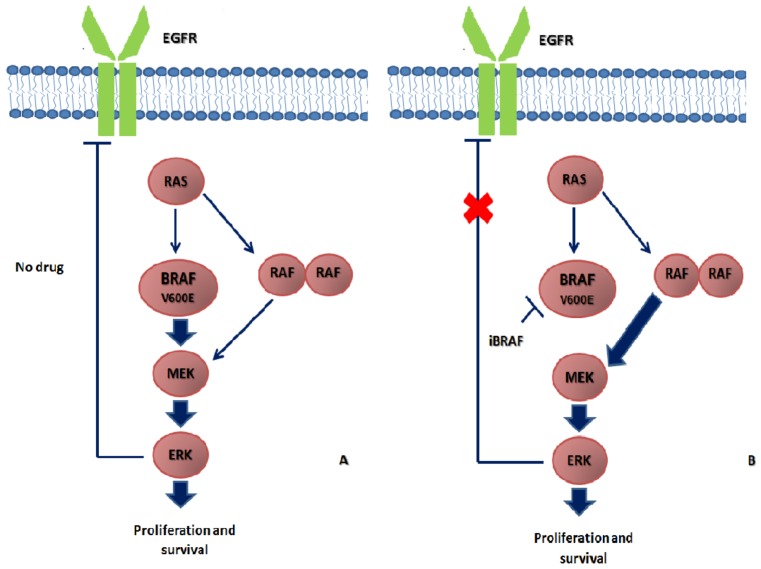
Adaptive feedback signaling in BRAF V600E-mutated mCRC. (**A**) In BRAF V600E-mutated mCRC, activated BRAF V600E monomer activates the MAPK pathway (MEK and ERK), leading to cell growth. Activated ERK suppresses the upstream activation of MAPK pathway through negative feedback on TRK, such as EGFR. (**B**) BRAF inhibitors (iBRAF) lead to transient inhibition of MAPK pathway and loss of ERK-dependent negative feedback on RTK, resulting in paradoxical activation of MAPK pathway.

**Table 1 ijms-20-05369-t001:** Main clinical trials with BRAF-inhibitors in BRAF-mutated mCRC.

	Treatment	N° of Patients	RR (%)	mPFS (months)	mOS (months)
	Vemurafenib [62]	21	5%	2.1 m	7.7 m
Single BRAF inhibitor	Vemurafenib [60]	10	0%	4.5 m	9.3 m
	Encorafenib [82]	18	0%	4 m	-
	Vemurafenib + cetuximab [60]	27	3.7%	3.7 m	7.1 m
Doublet BRAF + EGFR inhibitor	Vemurafenib + panitumumab [72]	15	13%	3.2 m	7.6 m
	Dabrafenib + panitumumab [75]	20	10%	3.5 m	-
	Encorafenib + cetuximab [74]	50	22%	4.2 m	-
Doublet BRAF + MEK inhibitor	Dabrafenib + trametinib [76]	43	12%	3.5 m	-
Triplet BRAF + MEK + EGFR inhibitors	Dabrafenib + trametinib + panitumumab [75]	91	21%	4.2 m	9.1 m
Triplet BRAF + MEK + EGFR inhibitors	Encorafenib + cetuximab +/− binimetinib [81]	224 (triplet) 220 (doublet)	26% 20%	4.3 4.2	9.0 m 8.4 m
Triplet BRAF + EGFR + PI3K inhibitors	Encorafenib + cetuximab + alpelisib [74]	52	27%	5.4 m	15.2 m
Triplet BRAF + EGFR inhibitors + irinotecan	Vemurafenib + cetuximab + irinotecan [80]	106	16%	4.4 m	-

RR: response rate; mPFS: median progression free survival; mOS: median overall survival; m: months.

**Table 2 ijms-20-05369-t002:** Ongoing studies in BRAF-mutated mCRC.

Therapeutic Strategy	ClinicalTrials Identifier	Agents Investigated	Phase	Status
BRAF + EGFR + MEK inhibition	NCT01750918 [87]	Dabrafenib + panitumumab vs. dabrafenib + trametinib + panitumumab Dabrafenib + panitumumab vs. dabrafenib + trametinib + panitumumab vs. 5-FU-based chemotherapy + monoclonal antibody	II, open label	Active, not recruiting
Chemotherapy + selective Wee-1 inhibitor	NCT02906059 [88]	Irinotecan + AZD1775	Ib	Active, recruiting
BRAF + MEK + CDK4/CDK6 inhibitor	NCT01543698 [89]	Binimetinib + encorafenib vs. binimetinib + encorafenib + LEE011	Ib/II	Active, not recruiting
PORCN (Wnt-pathway) inhibitor + immunotherapy	NCT01351103 [90]	LGK974 +/− PDR001	I	Active, recruiting
BRAF + EGFR + Wnt pathway inhibitor	NCT02278133 [84]	Encorafenib + cetuximab + WNT974	I/II	Not active, not recruiting
BRAF + EGFR inhibitor + chemotherapy	NCT03727763 [91]	FOLFIRI + cetuximab + vemurafenib	II	Active, recruiting

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
