# Peer review of "BRAF-Mutated Colorectal Cancer: Clinical and Molecular Insights"

_ijms, 2019, doi:10.3390/ijms20215369_

Round 1

Reviewer 1 Report

This is a comprehensive review on BRAF mutation (V600E) in colorectal cancers (CRC) which is observed in ~10% of CRC cases, and how it affects prognosis and treatment with various monoclonal antibodies (Bevazizumab) and inhibitors (FOLFOXIRI).  The authors also discussed in detail about the BEACON trial (BRAF/MEK combination therapy) used in CRC treatment.  The review provides detailed information on patient resistance to standard treatment of BRAF-mutated CRC, with a focus on molecular heterogeneity and on research perspectives on translational and clinical aspects.     

Reviewer 2 Report

The authors reviewed the heterogeneity of BRAF mutated CRC. They discussed the mechanisms for resistance of BRAFi, current clinic trials, and future therapeutic directions. There are a few minor concerns:

The title may need to be changed as it is somewhat misleading. Essentially the authors just want to emphasize the tumor heterogeneity, which is applicable to any kind of tumor. Line 48 and 83 are duplicate as they are talking about the same thing: BRAF mutation is found in about 10% mCRC. Line 108, please define “cycle 20 checkpoint associated genes”? For the BEACON study, it should be moved to the “Triplet combinations” section and table 1. The discussion about Ref 82 should also be moved to the “Triplet combinations” section. The significance of this study may need to be further explained as studies such as ref 74/75/77 are all have comparable or even better efficacy. Is the large cohort make it distinct?
